# Targeted Inhibition in Pediatric MET and ALK-Altered Hemispheric Gliomas: Objective Responses Followed by Treatment Resistance

**DOI:** 10.3390/ijms26209864

**Published:** 2025-10-10

**Authors:** David Wilson, Sateesh Jayappa, Lora Parker, Eylem Ocal, Tomoko Tanaka, Murat Gokden, Kevin Bielamowicz

**Affiliations:** 1Department of Pediatrics, University of Arkansas for Medical Sciences (UAMS), Little Rock, AR 72223, USA; 2Section of Pediatric Hematology/Oncology, University of Arkansas for Medical Sciences (UAMS), Little Rock, AR 72223, USA; 3Arkansas Children’s Hospital (ACH), Little Rock, AR 72223, USA; 4Department of Radiology, University of Arkansas for Medical Sciences (UAMS), Little Rock, AR 72223, USA; 5Department of Neurosurgery, University of Arkansas for Medical Sciences (UAMS), Little Rock, AR 72223, USA; 6Department of Pathology, University of Arkansas for Medical Sciences (UAMS), Little Rock, AR 72223, USA; 7Division of Pediatrics, The University of Arkansas for Medical Sciences (UAMS), 1 Children’s Way Slot 512-10, Little Rock, AR 72223, USA

**Keywords:** brain tumor, pediatric high-grade glioma, glioma, MET alteration, ALK alteration, pediatric CNS tumor

## Abstract

Pediatric-type diffuse high-grade gliomas (pHGGs) tend to have a dismal prognosis. Some of these gliomas feature alterations in genes such as ROS1, ALK, MET, and NTRK1–3. Despite development of targeted agents, the therapeutic application of these agents in pHGGs is still unclear. The aim of this retrospective case series is to report the outcome of two patients with pHGGs who were treated at Arkansas Children’s Hospital with targeted agents (Cabozantinib for a MET fusion in patient 1 and Lorlatinib for an ALK fusion in patient 2) with an initial, objective response followed by treatment resistance. Each diagnosis was determined based on histology, targeted tumor sequencing, and methylation profiling. In both cases, relapse occurred while on targeted inhibition. Recurrent tumor sequencing for patient 2 revealed a MET copy gain suggesting a mechanism of resistance in this patient. Pediatric high-grade gliomas with targetable alterations can show objective responses to pathway inhibition. Relapse after initial response may warrant additional surgical samples to identify new alterations which can lead to changes in therapy. Larger prospective cohorts are needed to study targeted agents in this population, and earlier integration of these agents may be beneficial.

## 1. Introduction

Central Nervous System (CNS) tumors are the most common cause of cancer-related deaths in children [1]. Pediatric-type diffuse high-grade gliomas (pHGGs) are the most common malignancy of the CNS in childhood, and the average survival rate in children at 3 years is approximately 30% [2]. The World Health Organization’s (WHO) fifth and latest iteration of classification for CNS tumors divides pHGGs into four categories: K27 M-altered, H3 G34-mutant, H3/IDH wild-type and infant-type hemispheric glioma (IHG) [3]. Regardless of category, the aggressive nature of pHGGs has prompted a multi-modal approach to treatment.

Preliminary results from a study published in 2005 established surgical resection, radiotherapy and Temozolomide (TMZ) as the standard of care for adult high-grade gliomas. The long-term results of the study showed that TMZ, which is an alkylating agent, was particularly effective in adult patients with methylated MGMT promoter [4]. This strategy, when applied to pHGGs, did not improve outcomes as demonstrated in a Children’s Oncology Group (COG) trial (ACNS0126) [5]. A subsequent COG trial (ACNS0423) combined TMZ with Lomustine as maintenance therapy following TMZ/radiation; this combination showed improved Event-Free Survival (EFS) and Overall Survival (OS), especially in patients with MGMT overexpression compared to historical cohorts [6]. There have been other chemotherapy strategies utilized in younger children with high-grade glioma (HGG) such as BABY POG and European HGG protocols; in general, these patients have had better survival outcomes compared to older children with HGGs, but the best use of adjuvant therapy in this population is not clear [7].

High-grade gliomas in infants and young children have shown some response to chemotherapy, but methods of achieving local disease control, including surgery and radiation, can be associated with high morbidity [8]. The availability of molecular next-generation sequencing (NGS) has made it possible to identify potential therapeutic targets for children with pHGGs. One example of this is identification of HGGs with BRAFV^600e^ alterations, which activates the mitogen-activated protein kinase pathway (MAPK). BRAF^V600e^ mutant HGG only accounts for roughly 5–10% of pHGGs. A Phase 2 study using the BRAF-inhibitor Dabrafenib and the MEK-inhibitor Trametinib to treat relapsed or refractory pHGGs produced a median overall survival of 32.8 months [9]. Based on these results, the Food and Drug Administration in June 2022 granted accelerated approval to this combination for the treatment of adults and children over the age of 5 years, for treatment of refractory/relapsed HGG. Promising results from a pilot study using these two drugs for upfront therapy [10], has spurred a larger, multi-institutional Phase 2 study from the Children’s Oncology Group (ACNS1723) that is currently ongoing to study the utility of utilizing targeted agents upfront in BRAF^V600e^ mutant pHGGs.

There is some evidence that patients with pHGGs that harbor other Receptor Tyrosine Kinase (RTK) pathway alterations respond favorably to targeted agents [11,12]. One patient with IHG and ALK fusion experienced a dramatic response to the ALK inhibitor Lorlatinib [13]. Another patient experienced a clinical and radiographic response when Larotrectinib, a pan-NTRK inhibitor, was used at relapse [14]. An 11-month-old female with IHG and NTRK fusion, confirmed via biopsy, was treated with upfront Larotrectinib monotherapy and experienced long-term stability of her tumor [15]. The patient started Larotrectinib 24 days after biopsy and had stable disease for several years with no progression [15]. There have already been numerous clinical trials to assess various RTK-targeted therapies. There is a need to continue to identify more effective upfront treatments for this disease, as relapsed pHGGs carry a devastating prognosis in pediatric patients. The rate of progression-free survival for recurrent pHGGs is approximately 3.5 months [16].

In this study we present two patients from a single center with pHGGs that experienced dramatic responses to targeted agents prior to recurrence. Each of the cases featured a targetable RTK alteration: one was an IHG with MET alteration, and the other was a diffuse pediatric-type HGG MYCN subtype.

## 2. Case Description

Patient 1

A previously healthy 11-month-old male presented to the Emergency Department with increased fussiness, vomiting, hypotonia, developmental regression, and macrocephaly over the preceding several weeks. A head computed tomography (CT) scan showed a large, solid/cystic mass with calcifications in the left frontal lobe that occupied most of the left hemisphere (as shown in Figure 1A). The patient was subsequently admitted to the pediatric intensive care unit (PICU) and magnetic resonance imaging (MRI) scan of brain and spine confirmed a large, supratentorial, left-hemispheric mass with midline shift. There was no evidence of metastatic spread. An extra ventricular drain was placed at bedside on hospital day 1 due to worsening symptoms related to increased intracranial pressure. He underwent near-total resection of the tumor on hospital day 2. Histology of the tumor was consistent with a high-grade glioma with brisk mitotic activity, dystrophic calcifications, necrosis, vascular proliferation, and hypercellularity in a myxoid background (as shown in Figure 2). Immunohistochemically, it was positive for Vimentin, GFAP, olig2, S-100 and synaptophysin was negative with wild-type pattern of staining for p53. The Ki-67 proliferation index was 40–50%. DNA methylation profiling from the National Cancer Institute (NCI) revealed an integrated diagnosis of infant-type hemispheric glioma, WHO-CNS grade 4. Targeted sequencing showed a frameshift alteration in CHEK2 as well as CDKN2A and CDKN2B loss, MTAP loss and a CLIP2-MET fusion. MGMT promoter methylation was not present. Radiation therapy was deferred due to the patient’s age and location of the tumor. He was started on systemic chemotherapy via Pediatric Oncology Group 9233/34 [17] with Cyclophosphamide, Vincristine, Cisplatinum and Etoposide. His hospital course was complicated by worsening hydrocephalus requiring placement of ventriculo-peritoneal shunt on hospital day 22. He also developed culture negative meningitis, which delayed the start of chemotherapy. A subsequent MRI 4 weeks post resection and prior to starting chemotherapy showed progressive leptomeningeal disease. A repeat MRI obtained following the first cycle of chemotherapy showed progressive disease. Given the previously identified MET fusion, Cabozantinib, a MET inhibitor, at a dose of 40 mg/m^2^/day was added at the start of cycle two [18]. This dose matched the recommended phase 2 dose as defined in Children’s Oncology Group clinical trial ADVL1211 [18]. The Cabozantinib was administered enterically by crushing the tablet and dissolving in a small volume of water. The patient did not experience any systemic adverse effects as defined by the Common Terminology Criteria for Adverse Events (CTCAE). A repeat MRI Brain approximately 4 weeks later demonstrated a clear treatment response (as shown in Figure 1E) that met partial response criteria by the Response Assessment in Neuro-Oncology (RANO) criteria. A subsequent scan 4 weeks later showed clear progression, and the patient was discharged home on hospice. He ultimately succumbed to his illness approximately 4 months after his initial diagnosis.

Patient 2

A previously healthy 2-year-old female presented to the Emergency Department for vomiting, somnolence, and difficulty ambulating over the prior 2 weeks. MRI Brain re-demonstrated a 5.6 cm × 7 cm × 5.8 cm, heterogeneous solid/cystic enhancing tumor in the left frontoparietal lobe (as shown in Figure 3A). On hospital day 2, she underwent gross total resection of the mass. Histological examination showed a high-grade glial tumor, with brisk mitotic activity (as shown in Figure 4). A broad panel of immunochemistry stains showed it to be positive for olig2, GFAP, S-100 protein. Synaptophysin was negative and p53 was diffusely and strongly overexpressed, consistent with a TP53 alteration. Next-generation sequencing of the tumor tissue revealed an MBOAT2-ALK chromosomal rearrangement with TP53 loss of function and MYCN copy gain. DNA methylation profiling of the tumor was performed, which suggested the diagnosis of diffuse pediatric-type high grade glioma, MYCN subtype.

While waiting to begin focal external beam radiation therapy (XRT), she experienced recrudescence of symptoms. Approximately 4 weeks following resection, she presented to the ED with lethargy, confusion and new onset seizure-like activity. A repeat brain MRI revealed recurrence of the tumor in the surgical site bed. She started focal XRT with concurrent temozolomide. She received 5400 cGy over 30 fractions of focal proton therapy. A subsequent MRI obtained 4 weeks after radiation demonstrated clear progression of disease (as shown in Figure 4). Approximately 1 week after that scan she started the ALK inhibitor Lorlatinib at a dose of 92 mg/m^2^/day. The drug was administered orally by crushing the tablet and mixing in a small volume of liquid so the patient could take it orally. The patient did not experience any systemic adverse effects as defined by the CTCAE. A repeat MRI Brain 2 months after starting Lorlatinib showed a clear decrease in the size of the recurrent lesions (as shown in Figure 3D), with further ongoing response 4 months after treatment initiation (as shown in Figure 3E) which met RANO criteria for a partial response. Unfortunately, 6 months after starting Lorlatinib, scans showed recurrence of disease both inside the radiation field focally and outside the radiation field distantly (as shown in Figure 3F). She underwent subsequent resection of two of the disease foci, which she tolerated well. While the histology of the recurrent tumors was similar (shown in Figure 4), molecular sequencing again confirmed the MBOAT2-ALK fusion and MYCN amplification, but also revealed a new MET copy gain. Lorlatinib was restarted after being held during the perioperative period. In addition, she was started on Bevacizumab every 2 weeks along with daily Cabozantinib. While the combination was tolerated well, scans 8 weeks after beginning this regimen showed further progression of disease. She was readmitted shortly after this and found to be obtunded. She quickly developed progressive symptoms at that point and died approximately 15 months after diagnosis.

## 3. Discussion

In this case series we present two examples of patients with pHGGs exhibiting alterations that experienced clear objective responses when treated with corresponding targeted agents. Patient 1 had an IHG with a MET fusion and near-complete response when treated with Cabozantinib, a MET inhibitor, with concurrent systemic chemotherapy after progression on traditional chemotherapy alone. Traditional chemotherapy may have contributed to the disease response for patient 1. The leptomeningeal deposits intensified after the first cycle of chemotherapy alone (as shown in Figure 1D) and there was a near-complete response in those deposits following the addition of Cabonzantinib to the second cycle of chemotherapy (as shown in Figure 1E), suggesting that the addition of the targeted agent played a primary role in this response. While this cannot be surely known, the response itself followed by subsequent recurrence may warrant further investigation. Patient 1 is classified by methylation profiling as IHG. IHGs are defined by an astrocytic morphology, presentation in early childhood, hemispheric location and usually feature Receptor Tyrosine Kinase (RTK) alterations in genes such as ROS1, ALK, MET and NTRK [3,19,20]. Patient 2’s case was less clear despite having an ALK alteration typically seen in IHG, the hemispheric location, and young age at presentation. The tumor showed N-MYC copy number gain but no alterations in EGFR as often seen in N-MYC subtype pediatric HGG.

Patient 2 initially had an ALK fusion and had a near-complete response to the ALK inhibitor Lorlatinib after progression post radiation with concurrent Temozolomide. The recurrent tumor specimen also showed an acquired MET copy gain (as shown in Table 1). While this acquired change alone may not completely explain the mechanism of resistance to Lorlatinib, it is clear in this case new alterations were present at relapse and may have contributed to the aggressive clinical course at relapse in patient 2. She was also treated with Cabozantinib following second resection, but her disease ultimately progressed on this treatment. We believe the patient’s family was adherent to all medications. The mean OS for pHGGs with MYCN alteration is approximately 16.4 months [21]. The acquired alteration highlights the importance of repeating a biopsy at the time of recurrence.

A recent cohort of IHG patients at one institution who were treated with surgery and chemotherapy experienced more than 90% survival rate at 5 years, but they experienced significant neurocognitive decline as a result of treatment [8]. In this study, more than 70% of the patients had pathogenic gene fusions involving the RTK pathway. Interestingly, extent of resection did not have any significant bearing on prognosis in that study [8]. Upfront resection of any kind is not always feasible in these patients as they are typically very young, have less reserve, and are more susceptible to complications from blood loss. The relative size of the tumor may contort brain vasculature leaving the patient vulnerable to those complications [22]. A similar case involved a 3-year-old boy whose tumor could not be fully resected due to life-threatening hemorrhage, but an ALK fusion was identified after partial resection, and he was started on Lorlatinib. He experienced a dramatic response both clinically and radiographically, eventually allowing for a safe total resection [23].

Although both patients in this case series ultimately experienced poor outcomes, they offer meaningful contributions to the growing body of evidence for potential treatments in these rare tumors. Neither of the patients experienced any adverse effects from their respective targeted agents, and quality of life was not evaluated in this limited study. This case series contains a limited sample size from a single institution, the cases represent the ability for pHGGs with RTK alterations to respond to targeted agents but also development of resistance mechanisms which likely limit these treatments [24,25]. Resistance mechanisms have already been described in other diseases such as lung cancer, where acquired resistance occurs following treatment with targeted agents [26]. While it is unclear why resistance occurred in these patients, future studies are needed to better understand these mechanisms. Confirmation of these findings with a larger multi-center retrospective study is needed. Prospective studies integrating targeted agents in IHG patients earlier in treatment may be warranted.

## 4. Methods

Two pediatric patients with pHGGs treated at Arkansas Children’s Hospital (ACH) are described. The research project was submitted to and approved by the institutional review board (IRB) of the University of Arkansas for Medical Sciences (UAMS) on 18 September 2024 (Approval code: FWA00001119) and was determined to be exempt from consent and full IRB review. Medical information for the patients was obtained from the electronic medical records at ACH. Tumor sequencing was performed by Tempus xT 648 gene targeted sequencing panel, including both somatic and germline sequencing (www.tempus.com), accessed on 30 July 2025. Methylation profiling for all three specimens was performed by the Laboratory of Pathology at the National Cancer Institute/National Institutes of Health, using versions 11b6 and 12b6 of the Heidelberg classifier and the NCI/Bethesda classifier.

## Figures and Tables

**Figure 1 ijms-26-09864-f001:**
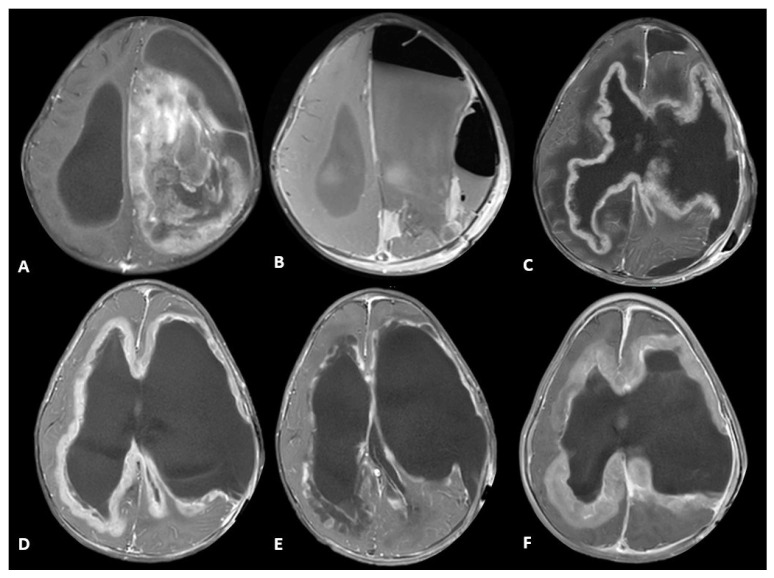
**Series of axial T1-weighted MR images post-contrast for patient 1.** (**A**) Initial imaging from diagnosis demonstrates a large left frontoparietal complex solid and cystic mass with some punctate calcifications; (**B**) follow-up imaging from post-operative day 1 shows near-total resection; (**C**) follow-up scan approximately 1 month post-op demonstrates clear progression with diffuse, enhancing leptomeningeal deposits; (**D**) follow up MRI performed 2 months post-op and following 1 cycle of chemotherapy shows stable to progressive disease; (**E**) follow-up scan approximately 3 months post-op following the addition of Cabozantinib shows improvement and near-absence of previous leptomeningeal disease; (**F**) follow-up scan approximately 4 months post-op and following 2 cycles of chemotherapy and 2 months of Cabozantinib shows clear recurrence of leptomeningeal disease.

**Figure 2 ijms-26-09864-f002:**
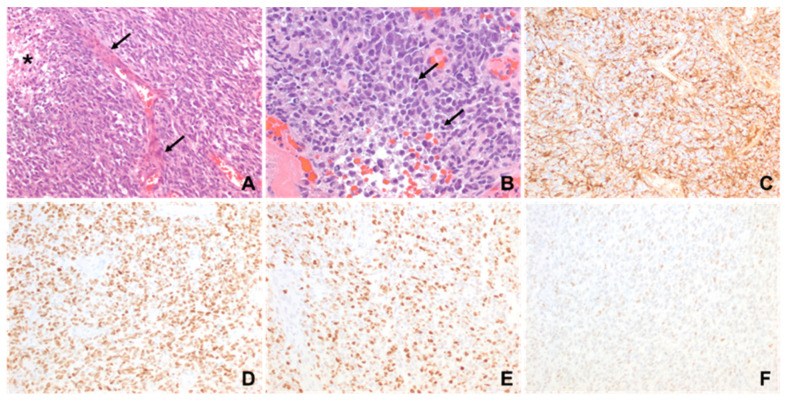
Microscopic features of the resection specimen in 1-year-old male (patient 1). (**A**) A hypercellular glial neoplasm with spindled nuclei, vascular proliferation (arrows) and necrosis (*) is seen. (**B**) Frequent mitotic figures (arrows) are present. (**C**) GFAP and (**D**) olig2 are positive. (**E**) Proliferation index is high. (**F**) P53 shows wild-type pattern (original magnifications: (**A**,**C**–**F**,) 200×; (**B**), 400×).

**Figure 3 ijms-26-09864-f003:**
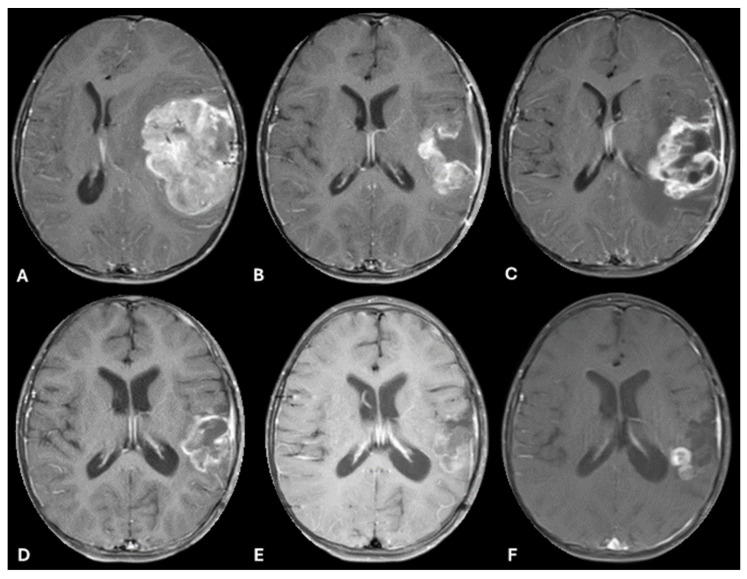
A 2-year-old female (patient 2) with MRI T1-weighted, post-contrast, axial images. (**A**) Shows a large, left cerebral hemispheric heterogenous mass in the temporal lobe at diagnosis; (**B**) 6 weeks following diagnosis and resection, prior to starting chemo-radiation therapy, shows recurrent disease along the surgical cavity; (**C**) post-radiation and concurrent Temozolomide showing progressive disease; (**D**) 5 months after radiation and 2 months after starting Lorlatinib; (**E**) 4 months after starting Lorlatinib showing near-complete resolution with minimal enhancement along the surgical cavity; (**F**) 6 months after starting Lorlatinib showing new small, nodular enhancements outside the original surgical cavity and radiation field with leptomeningeal deposits representing a second recurrence.

**Figure 4 ijms-26-09864-f004:**
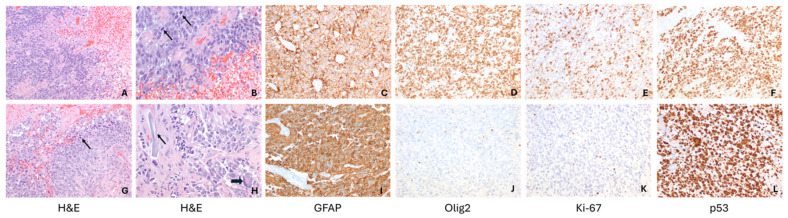
Comparative microscopic features of initial and recurrence resection specimens in a 2-year-old female (patient 2). (**A**–**F**) First resection: a neoplasm with small blue cell appearance (**A**,**B**) is seen, with necrosis ((**A**), **right upper corner**) and mitotic activity ((**B**), arrows). GFAP and olig2 positivity ((**C**) and (**D**), respectively) indicate its glial nature. Proliferation index is high (**E**), and p53 shows mutant pattern (**F**). (**G**–**L**) recurrence specimen: A neoplasm with small blue cell appearance (**G**,**H**) is again seen, with necrosis ((**G**), **upper left corner**) and mitotic activity ((**G**), arrow). Surgical material ((**B**), arrow), with associated histiocytic reaction, as evidence of previous surgery, and nuclear enlargement and multinucleation ((**B**), block arrow) as a result of radiation treatment are present. GFAP is again positive (**I**), but olig2 expression is lost (**J**). Proliferation index is significantly lower (**K**). P53 shows mutant pattern (**L**). (Original magnifications: (**A**,**C**–**G**,**I**–**L**) 200×; (**B**,**H**) 400×).

**Table 1 ijms-26-09864-t001:** Key highlights of patient cases.

	Tumor Sample	Histopathology	Targeted Sequencing	Methylation Profiling Classification	Integrated Diagnosis	Treatment Regimen
**Patient 1**	Diagnostic	-Brisk mitotic activity -Gliosis in myxoid background -Vimentin positive -olig2 positive -S-100 positive	-CLIP2-MET fusion -CHEK2 frameshift alteration -CDKN2A, CDKN2B, MTAP loss	Infant-type Hemispheric Glioma	Infant-type Hemispheric Glioma, WHO-CNS Grade 4	Systemic chemotherapy: Cyclophosphamide, Vincristine, Cisplatinum and Etoposide (2 cycles) Targeted agent: Cabozantinib
**Patient 2**	Diagnostic	-Brisk mitotic activity -INI1 retained -P53 diffusely expressed -olig2 positive	-MBOAT2-ALK fusion -TP53 loss -MYCN copy gain	Diffuse Pediatric-type High-Grade Glioma, MYCN subtype	Diffuse Pediatric-type High-Grade Glioma, H3-wild-type and IDH-wild-type, CNS WHO Grade 4	Radiotherapy with concurrent Temozolomide Targeted agent: Lorlatinib
	Relapse	Similar to diagnostic sample, with olig2 loss	-MET copy gain -MBOAT2-ALK fusion -TP53 loss -MYCN copy gain	Diffuse Pediatric-type High-Grade Glioma, MYCN subtype	Diffuse Pediatric-type High-Grade Glioma, H3-wild-type and IDH-wild-type, CNS WHO Grade 4	Target agents: Lorlatinib and Cabozantinib

## Data Availability

The original contributions presented in this study are included in the article. Further inquiries can be directed to the corresponding author.

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
