# Peer review of "Targeted Inhibition in Pediatric MET and ALK-Altered Hemispheric Gliomas: Objective Responses Followed by Treatment Resistance"

_ijms, 2025, doi:10.3390/ijms26209864_

Round 1

Reviewer 1 Report

Comments and Suggestions for Authors

The publication is interesting and valuable for the reader, especially in the context of future research on high-grade gliomas. It does not contribute substantially to the current literature due to the inclusion of only two patients; however, even a few case reports can help build a foundation for future treatment modifications.

Author Response

Reviewer comments: The publication is interesting and valuable for the reader, especially in the context of future research on high-grade gliomas. It does not contribute substantially to the current literature due to the inclusion of only two patients; however, even a few case reports can help build a foundation for future treatment modifications.

Author response: Thank you very much for your thorough review of the manuscript. We agree that this adds to the existing literature and will continue to build on a foundation for these rare tumor types.

Reviewer 2 Report

Comments and Suggestions for Authors

The authors provide a thorough report of two cases of pHGG and insights into targeted therapy resistance in pHGG. There are some suggestions.

Figures 1 and 3: Arrows can be added to highlight the key changes of the tumor mass.

When Figure 3 is mentioned in the text, please specify the panel(s) that the authors are referring to.

As the authors are investigating the resistance to therapeutics, it would be helpful to add the treatment regimens in Table 1.

Author Response

Reviewer 2 comment 1: The authors provide a thorough report of two cases of pHGG and insights into targeted therapy resistance in pHGG. There are some suggestions.

Author response: Thank you very much for your thorough review of our manuscript. 

 Reviewer 2 comment 2: Figures 1 and 3: Arrows can be added to highlight the key changes of the tumor mass.

Author response: Thank you very much for your thorough review of our manuscript. We considered the addition of these arrows carefully but ultimately felt that it cluttered the figures too much leading to confusion for the reader.

Reviewer 2 comment 3: When Figure 3 is mentioned in the text, please specify the panel(s) that the authors are referring to.

Author response: Thank you very much for this suggestion. We agree this brings increased clarity to the reader to include this information in the body of the text. We have made the necessary adjustments to include specific panels being referenced as shown in Lines 197, 198, 200.

Reviewer 2 comment 4: As the authors are investigating the resistance to therapeutics, it would be helpful to add the treatment regimens in Table 1.

Author response: Thank you for the suggestion. We have incorporated the treatment regimens for each patient into the table.

Reviewer 3 Report

Comments and Suggestions for Authors

This is a paper about description of objective responses of targeted therapy on pHGG cases .

Major concerns

Patient 1: After 1 cycle of chemotherapy authors stated the presence of stable to progressive disease. As the fluid part increased inside the tumor, not the wall (according to Fig 1D) , it is really hard to say that this is a real progression. If different chemotherapy was not given in further cycles, I assume that it was authors' opinion, too. Therefore, further chemotherapy could have had additional  positive effect (clear regression). How is it surely known that adding cabozantinib resulted in the additional positive effect?

Patient 2: This is not a clear case for lorlatinib effect. According to the case description 4 weeks after irradiation there was a "clear progression", which resulted in  start of ALK-inhbitor. However, based on Fig 3c and possible known consequences of irradiation it could be a pseudoprogression after irradiation only and not a real progression. Similarly, the state of the tumor on Fig 3D could be the consequence the effect of irradiation and  TMZ after diminishing of post irradiation pseudo-progression. Further effect could be attributed to lorlatinib effect, but this phenomenon was already described by Bagchi (N Engl J Med 2021;385:761-763), which paper is not cited by authors.

Author Response

Reviewer 3 Comment 1: (Patient 1) After 1 cycle of chemotherapy authors stated the presence of stable to progressive disease. As the fluid part increased inside the tumor, not the wall (according to Fig 1D) , it is really hard to say that this is a real progression. If different chemotherapy was not given in further cycles, I assume that it was authors' opinion, too. Therefore, further chemotherapy could have had additional  positive effect (clear regression). How is it surely known that adding cabozantinib resulted in the additional positive effect?

Author response: Thank you very much for this suggestion. The patient experienced clear progression following resection and prior to chemotherapy as shown in Figure 1C. While it is less clear that the patient experienced progression in Figure 1D compared to 1C, it is difficult to capture the entirety of an MRI in a single axial plane, however, for the ease of presentation this was the best section to represent all of the imaging timepoints. Our Neuroradiology team felt at this timepoint there was clear progression of the disease. Regardless, we changed our wording to reflect this as shown beginning in Line 238: "The leptomeningeal deposits intensified after the first cycle of chemotherapy alone (as shown in Figure 1D) and there was a near-complete response in those deposits following the addition of Cabozantinib to the second cycle of chemotherapy (as shown in Figure 1E), suggesting that the addition of the targeted agent played a primary role in this response. While this cannot be surely known, the response itself followed by subsequent recurrence may warrant further investigation."

Reviewer 3 Comment 2: (Patient 2) This is not a clear case for lorlatinib effect. According to the case description 4 weeks after irradiation there was a "clear progression", which resulted in  start of ALK-inhbitor. However, based on Fig 3c and possible known consequences of irradiation it could be a pseudoprogression after irradiation only and not a real progression. Similarly, the state of the tumor on Fig 3D could be the consequence the effect of irradiation and  TMZ after diminishing of post irradiation pseudo-progression. Further effect could be attributed to lorlatinib effect, but this phenomenon was already described by Bagchi (N Engl J Med 2021;385:761-763), which paper is not cited by authors.

Thank you for this suggestion. We agree there are confounding factors to a suggestion that the response is related to Lorlatinib. These cases pose intriguing potential based on the responses, though. The potential represented in the responses of these two patients warrants further investigation in the future. On review of the case our Neuroradiology team favored true progression over pseudo-progression given the appearance of the tumor and MRI sequences performed.

We agree that the phenomenon described by Bagchi and others was an important supporting case and for this reason we included a description of the case as well as the reference in the original manuscript, as stated in Lines 288-291. This was already included prior to this comment.

Reviewer 4 Report

Comments and Suggestions for Authors

Thank you for submitting this interesting case series on pediatric high-grade gliomas with MET and ALK alterations. The cases are clearly described, supported by molecular profiling and imaging, and the message regarding initial response followed by resistance is clinically valuable.

A few points require clarification or improvement before publication:

  1. Clarify which radiographic response criteria were used (RANO/mRANO or volumetric) when reporting “objective response.”

  2. Provide systematic toxicity data (per CTCAE) for cabozantinib and lorlatinib, and acknowledge that the cabozantinib dose aligns with the pediatric RP2D (COG ADVL1211).

  3. Correct terminology: replace “acquired MET fusion” with MET copy gain for consistency.

  4. In the Discussion, expand on resistance mechanisms (ALK bypass via MET, as reported in neuroblastoma and lung cancer literature).

  5. Refine the Conclusions: highlight the importance of re-biopsy at relapse and clarify that quality of life was not assessed in this study.

These are minor revisions. Addressing them will significantly strengthen the manuscript.

Author Response

Reviewer 4 Comment 1:  Clarify which radiographic response criteria were used (RANO/mRANO or volumetric) when reporting “objective response.”

Author's response: Thank you for this suggestion. We did not utilize a radiographic response criteria, initially. We have changed our wording in the text to more accurately describe the nature of the response to: "clear treatment response." We also reassessed the images and found that both cases met partial response by RANO criteria. So we have added this to the body of the text as well as shown on Line 150: “…that met partial response criteria by the Response Assessment in Neuro-Oncology (RANO) criteria.” As well as Line 203: “…which met RANO criteria for a partial response.”

Reviewer 4 Comment 2: Provide systematic toxicity data (per CTCAE) for cabozantinib and lorlatinib, and acknowledge that the cabozantinib dose aligns with the pediatric RP2D (COG ADVL1211).

Thank you for this suggestion. We agree these changes would make the paper stronger and we have added the following information to the manuscript:

Line 144-145: “This dose matched the recommended phase 2 dose as defined in Children’s Oncology Group clinical trial ADVL1211.”

Line 147-148: “The patient did not experience any systemic adverse effects as defined by the Common Terminology Criteria for Adverse Events (CTCAE).”

Line 199: “The patient did not experience any systemic adverse effects as defined by the CTCAE.”

Line 290-291 of the Discussion: “Neither of the patients experienced any adverse effects from their respective targeted agents and quality of life was not evaluated in this limited study.”

 Reviewer 4 Comment 3: Correct terminology: replace “acquired MET fusion” with MET copy gain for consistency.

Thank you for this suggestion. We have adjusted the wording in the discussion to reflect this (Line 252).

Reviewer 4 Comment 4: In the Discussion, expand on resistance mechanisms (ALK bypass via MET, as reported in neuroblastoma and lung cancer literature).

Thank you for this suggestion. We agree that this would add to the strength of the manuscript. We have added the following as shown in Line 299: “Resistance mechanisms have already been described in other diseases such as lung cancer, where acquired resistance occurs following treatment with targeted agents. While it’s unclear why resistance occurred in theses patients, future studies are needed to better understand these mechanisms.”

Reviewer 4 Comment 5: Refine the Conclusions: highlight the importance of re-biopsy at relapse and clarify that quality of life was not assessed in this study.

Author response: Thank you for this suggestion. We agree with adding both of these points to the manuscript and we have adjusted the language, as below:

Line 278: “The acquired alteration highlights the importance of repeating a biopsy at the time of recurrence.”

Line 294-295: “Neither of the patients experienced any adverse effects from their respective targeted agents and quality of life was not evaluated in this limited study.”